# Successful Use of Multidisciplinary Palliative Care in the Outpatient Treatment of Disseminated Histoplasmosis in an HIV Positive Child

**DOI:** 10.3390/children8040273

**Published:** 2021-04-02

**Authors:** Alison Lopez, Jason Bacha, Carrie Kovarik, Liane Campbell

**Affiliations:** 1Pediatric Infectious Diseases, University of Manitoba, Winnipeg, MB R3E3P5, Canada; alison.lopez17@gmail.com; 2Baylor College of Medicine International Pediatric AIDS Initiative at Texas Children’s Hospital, Houston, TX 77030, USA; bacha@bcm.edu; 3Department of Pediatrics, Baylor College of Medicine, Houston, TX 77030, USA; 4Baylor College of Medicine Children’s Foundation—Tanzania, Mbeya 53107, Tanzania; 5Department of Dermatology, Hospital of the University of Pennsylvania, Philadelphia, PA 19104, USA; carrie.kovarik@pennmedicine.upenn.edu

**Keywords:** pediatrics, global health, histoplasmosis, symptom management, HIV/acquired immunodeficiency syndrome (AIDS), resiliency

## Abstract

Histoplasmosis is an uncommon opportunistic infection in human immunodeficiency virus (HIV) positive children. The most common form is primary disseminated histoplasmosis, characterized by persistent fever and failure to thrive. A 10-year-old HIV positive girl presented to the Baylor College of Medicine Children’s Foundation—Tanzania Mbeya Center of Excellence (COE) with ulcerated skin lesions and a violaceous facial rash. She also had persistent fevers, severe acute malnutrition, and severe anemia. At diagnosis, the patient was failing first line antiretroviral therapy (ART) with a cluster of differentiation 4 immune cells (CD4) of 24 cells/µL and an HIV viral load (VL) of 196,658 cp/mL. The patient was changed to a second line ART regimen (abacavir, lamivudine, and ritonavir-boosted lopinavir) and received nutritional support, blood transfusions, multiple antibiotics, and meticulous wound care. She also received comprehensive symptom management, psychosocial support, and emergency housing through the COE’s palliative care program. Biopsy of a lesion showed intracytoplasmic organisms consistent with *Histoplasmosis capsulatum* var *capsulatum.* The patient was treated with conventional amphotericin B and oral itraconazole and she achieved wound healing as well as immune reconstitution and HIV viral suppression. Amphotericin infusions were given as an outpatient despite the resource constraints of the setting in southwestern Tanzania. Histoplasmosis should be considered in the differential diagnosis of the immunocompromised host with unusual skin manifestations and persistent fever.

## 1. Introduction

Histoplasmosis is a serious—yet uncommonly reported—opportunistic infection in children and adolescents living with human immunodeficiency virus (HIV) (CALHIV) in sub-Saharan Africa (SSA). The true burden of disease is not well known and is likely under-reported due to under-recognition of the disease by clinicians, difficulty in establishing a definitive diagnosis in settings with limited laboratory capability, and clinical overlap with other more common opportunistic infections such as tuberculosis [1]. Such under-recognition is of particular concern, as progressive disseminated histoplasmosis (PDH)—the most common presentation of the histoplasmosis in CALHIV—is fatal without appropriate treatment [2]. The symptoms of PDH in CALHIV are broad and nonspecific, including prolonged fever and failure to thrive, cough and difficulty breathing, hepatomegaly and splenomegaly, cytopenias and coagulopathy, diarrhea and/or cutaneous lesions, making clinical diagnosis particularly challenging [3,4]. Severe immunosuppression is often present at diagnosis of PDH in CALHIV; a case series reported median cluster of differentiation 4 immune cells (CD4) counts of 39 cells/µL and 22 cells/µL in patients between the ages of 1 and 5 years and over 6 years, respectively [3]. Treatment of CALHIV with PDH is particularly complex due to drug–drug interactions between antifungal therapy and antiretroviral treatment [4], and poor availability of appropriate fungal therapy in many SSA settings [1]. Despite these challenges, we describe a case of an HIV positive child with PDH who was successfully treated with outpatient intravenous antifungal treatment and multidisciplinary palliative care.

## 2. Methods—Case Report

A 10-year-old HIV positive girl was admitted to the inpatient malnutrition ward in Mbeya, Tanzania in 2016 following a 2 month history of fevers, progressive ulcerated scalp lesions, indurated and violaceous facial plaques, and subcutaneous nodules on her neck, arms, and trunk (Figure 1a–c). The patient started antiretroviral therapy (ART) two years prior to presentation but reported poor adherence to her first line ART regimen of zidovudine, lamivudine, and efavirenz. At presentation, the patient had severe acute malnutrition, severe immunosuppression, and severe cytopenias (Table 1).

Due to her virologic and immunologic treatment failure, her ART regimen was switched to abacavir, lamivudine, and ritonavir-boosted lopinavir.

A wide range of possible diagnoses were considered based on the patient’s clinical presentation. A scalp biopsy was obtained on day 10 of admission of which the histologic analysis narrowed the focus to deep fungal infections, including histoplasmosis (both *Histoplasmosis capsulatum* var *capsulatum* and *Histoplasmosis dubuosii* (“African type”)) as well as other intracellular infections such as *Leishmaniasis*. Special stains were performed, and the organisms were highlighted using a Grocott-Gomori’s (or Gömöri) methenamine silver stain (GMS), focusing the diagnosis on fungal infections. Biopsy results showed a dense, interstitial infiltrate of histiocytes containing intracellular organisms consistent with *H. capsulatum* var *capsulatum* (Figure 2). On histopathology, small budding intracellular yeasts measuring 2–4 microns, characteristic of *Histoplasmosis capsulatum* var *capsulatum*, were visible. Given the size of the organisms and the appearance (intracellular—no organism over the size of ~4 microns), *Histoplasmosis dubuosii* (“African type”) was unlikely. *Histoplasmosis dubuosii* (“African type”) is characterized in tissue by lemon shaped budding yeasts of 7–12 microns, none of which were seen on the tissue sections [1]. No organism in the sections was over 4 microns. The main organism of differentiation was *Leishmanisis*. Under oil, there was no kinetoplast, and the organisms did not stain for GMS, confirming the diagnosis of *Histoplasmosis capsulatum* var *capsulatum.*

On day 18 after presentation, she received her first dose of intravenous (IV) amphotericin B deoxycholate as an inpatient, then was discharged to continue her daily amphotericin infusions, together with wound care, directly observed therapy (DOT), nutritional support, and other supportive care as an outpatient at the Baylor College of Medicine Children’s Foundation—Centre of Excellence (COE) in Mbeya, Tanzania.

As an outpatient, her daily amphotericin infusions were slowly uptitrated to 1 mg/kg per day, which she tolerated well. With each infusion, she received pre- and post-intravenous hydration, pre-medications to prevent toxicity (meperidine, paracetamol, and chlorphenamine), and potassium and magnesium electrolyte replacements. She was assigned a dedicated clinic nurse to administer all medications and monitor vitals and clinical condition during each infusion. Through the COE’s palliative care program, she received pain and symptom management including meperidine as analgesia prior to dressing changes and to treat rigors associated with amphotericin infusions. She and her mother received psychological counseling and support from the COE’s counselors and social workers. She received a daily nutritious lunch while receiving amphotericin B deoxycholate infusions, clothing, and toiletries, and she had the opportunity to participate in wish making activities. Because she and her mother lived approximately 100 km from the COE, she and her mother were placed in emergency housing nearby the COE to facilitate her daily amphotericin infusions. All care and treatment including infusions of amphotericin B deoxycholate was provided free of charge to the patient and her family.

## 3. Case Report—Results

She received a total of 20 mg/kg of IV amphotericin B deoxycholate (23 doses) over a 4 week period without complications (Figure 1d–f). On day 49, she was then transitioned to oral itraconazole (200 mg/day) based on her clinical improvement and a repeat biopsy which demonstrated clearance of organisms. She continued her DOT at the clinic and took her itraconazole with a carbonated cola beverage to increase the absorption of the drug. Three weeks into her course of oral itraconazole (day 88), she experienced clinical relapse, and repeat histopathology showed recurrence of histoplasmosis. Her dose of itraconazole was increased to 300 mg/day, however, a week later, her relapse continued to worsen (Figure 1g–i), thus she was switched back to amphotericin B infusions on day 95 and received an additional 19 mg/kg of amphotericin over an 8 week period (to reach a lifetime cumulative dose of 39 mg/kg). During this course of amphotericin, she experienced rigors which were treated with meperidine and transient nephrotoxicity (peak creatinine of 97 µmol/L and urea of 9.1 mmol/L), which resolved after completion of the amphotericin. Her scalp wounds showed improvement after the additional 8 weeks of amphotericin infusions (day 148, Figure 1j–l), thus she was again transitioned to oral itraconazole (300 mg/day) and showed sustained clinical improvement after 4 additional months (day 271, Figure 1m–o).

After 18 months of oral itraconazole, she had complete resolution of her wounds, and her itraconazole was discontinued (Figure 3). At this point, she had achieved immune reconstitution (CD4 count of 628 cells/µL) and viral suppression. Since stopping the itraconazole, she remained well. Her ART regimen was subsequently switched to dolutegravir, abacavir, and lamivudine as part of a 2019 national guideline change. Four years after initial presentation, the patient had no symptom recurrence and has maintained virologic suppression and immune reconstitution (Table 2).

## 4. Discussion

We describe an unusual case of progressive disseminated histoplasmosis in an HIV positive child who was successfully treated via a multidisciplinary palliative care approach in an outpatient setting using a combination of IV amphotericin B deoxycholate, oral itraconazole, effective ART, and supportive measures. To our knowledge, this is the first reported case of histoplasmosis presenting as large ulcerating scalp wounds in an HIV positive child in Tanzania.

*Histoplasma capsulatum* is a dimorphic fungi that is endemic in tropical and subtropical areas around the world. This fungus thrives in soil contaminated with bird feces or bat guano. Among HIV positive individuals, it is one of the most common endemic mycoses. There are two pathogenic variants of *H. capsulatum*—*H. capsulatum* var *capsulatum* (Hcc) and *H. capsulatum* var *duboisii* (Hcd). Cases of Hcc are described worldwide but are highly prevalent in the Mississippi and the Ohio River valleys of the United States as well as Central and South America. Hcd (African histoplasmosis) is essentially confined to the African continent, specifically Central Africa, West Africa, and Madagascar. Both Hcc and Hcd have been reported in Tanzania [1]. The true burden of histoplasmosis in Africa is not fully known due to limited access to diagnostics, under-reporting, and under-recognition of the disease. Oladele et al. found 470 reported cases of histoplasmosis from the African continent over the past six decades [1]. A study evaluating hospitalized febrile patients in northern Tanzania identified nine cases of probable histoplasmosis based on antigen testing [7].

The main risk factor for acquiring histoplasmosis is living in or traveling to an endemic area [8]. Although the Tanzanian highlands are not described to be endemic for histoplasmosis, our patient would have been exposed to soil contaminated with chicken feces, as free-range chickens are common in rural Tanzania. The clinical manifestations depend on the histoplasmosis variant involved. Hcc infections in immunocompetent individuals are typically self-limiting pulmonary infections [8]. Hcd, in contrast, presents with a subacute infection affecting skin, lymph nodes, and bones [1]. For both variants, the severity of disease depends on the intensity of exposure, the strain specific virulence factors, and the host cellular immunity [9]. In individuals who are unable to mount a cell mediated immune response, such as those with HIV/AIDS, the infection can lead to PDH, commonly defined as clinical illness that does not improve after 3 weeks of observation and is associated with physical/radiographic or laboratory evidence of extrapulmonary involvement [10]. PDH is fatal if left untreated [4]. In HIV positive children, prolonged fever and failure to thrive are the most common presenting features of PDH [4]. In this case, the diagnosis of PDH was made based on the patient’s clinical features of prolonged fever, weight loss and severe acute malnutrition, severe cytopenias, chest radiographic abnormalities, and histopathologic confirmation of Hcc in cutaneous lesions in the context of an HIV infected child with AIDS.

The diagnosis of histoplasmosis was made based on histopathology of the patient’s skin biopsy through collaboration with the University of Pennsylvania Department of Dermatology’s Africa Teledermatology program (http://africa.telederm.org, accessed on 14 January 2016). Direct visualization of the organisms from sterile sites is indicative of an active infection. Hcc is an intracellular organism and is histologically distinct from Hcd—a larger extracellular fungus [1]. It is important to note that *Leishmania* species can resemble *H. capsulatum* on histopathology, however, our patient’s facial plaques were atypical for cutaneous leishmaniasis. Culturing *H. capsulatum* from bodily fluids or tissue specimens is the gold standard for diagnosis, but growth may take up to 6 weeks [11]. The specificity of culture is 100%, but sensitivity varies depending on disease syndrome, burden of infection, source, and number of specimens collected [1,8]. In PDH, bone marrow aspirates yield the highest proportion of positive cultures [11]. Antigen testing by enzyme immunoassay is non-invasive, confers high sensitivity, provides rapid results, and is useful in immunocompromised populations that may have false negative serologic tests [11]. Serologic testing that detects *Histoplasma*-specific antibody are positive in about 41–69% of HIV positive adults with PDH compared to 82% of immunocompetent adults with PDH [4]. However, antibodies are only detectable 4–8 weeks after the infection and may revert to negative after 12–18 months [11]. Serologic and antigen testing can cross react with *Blastomyces*, *Paracoccidioides*, and *Coccidioides* species [11]. Unfortunately, these diagnostic modalities (except for histopathology on formalin preserved skin biopsies) are not available in our setting due to high cost, trained technologists, and the need for a biosafety level three laboratory.

The National Institute of Health and Infectious Diseases Society of America recommends at least 2 weeks of liposomal amphotericin B followed by at least 12 months of oral itraconazole for moderate to severe PDH in both HIV positive adults and children [10]. Longer treatment courses may be necessary in children with primary immunodeficiencies, immunosuppression, or severe disease [10]. Lipid formulations of amphotericin B are costly and can be difficult to obtain in low- and middle-income country (LMIC) settings. For these reasons, our patient was treated with amphotericin B deoxycholate. Despite its less favorable side effect profile, supportive therapy and close monitoring allowed our patient to safely receive her treatment as an outpatient. Immune reconstitution inflammatory syndrome is not commonly associated with histoplasmosis and was not observed in our case [12]. Lifelong suppression with antifungal agents should be considered in cases of relapse despite appropriate treatment or with prolonged immunosuppression [4]. Fluconazole is less efficacious and has been associated with drug resistance [13]. Posaconazole and isovuconazole have been successfully used as salvage therapy [12,14,15,16] but are rarely available in most LMIC settings.

Although outpatient parental antimicrobial therapy (OPAT) is well established in many centers, parental antifungals are uncommonly administered for several reasons [17]. There is a paucity of experience with administering antifungals in this setting, especially with amphotericin B. The unfavorable toxicity profile requires more intensive clinical monitoring and management by an experienced clinician. A retrospective chart review by Malani et al. described significant adverse events associated with community-based amphotericin B infusions, namely nephrotoxicity, electrolyte imbalance, infusion reactions, and venous access device complications [18]. A study by van de Peppel et al. reported more favorable outcomes with outpatient intermittent dosing of liposomal amphotericin B [19]. However, the small study cohort and the exclusion of amphotericin B deoxycholate limit the extrapolation of their results. Another important challenge to providing antifungal OPAT is that invasive fungal diseases are more likely to occur in patients with significant comorbidities and result in a more severe illness.

In our patient’s case, the hospital where she was admitted was unable to provide the supportive care and monitoring that were necessary to safely administer amphotericin B infusions. Our outpatient multidisciplinary palliative care approach was able to ensure that she safely received the infusions through meticulous nursing care and administration of a package of supportive care, which included IV hydration, electrolyte replacement, and symptomatic treatment of infusion related side effects, despite the resource limitations of this setting and the challenges of providing OPAT. Supportive care protocols utilized in the provision of amphotericin for treatment of cryptococcal meningitis in LMIC settings were used to develop this patient’s supportive care plan [20,21]. Additionally, the clinic’s unique multidisciplinary palliative care program was able to provide comprehensive symptom management and psychosocial care to overcome the logistical and the socioeconomic barriers this young patient faced. Members of the palliative care program encouraged this patient to utilize resiliency to bravely endure months of intravenous amphotericin infusions and wound care. While there is a paucity of pediatric palliative care programs in Tanzania and in many LMICs [22,23], multidisciplinary palliative care can be an essential component in the management of pediatric and adolescent patients with HIV/AIDS and life threatening opportunistic infections [24]. In this case, outpatient administration of amphotericin allowed this patient to receive life-saving anti-fungal treatment and to achieve a sustained positive outcome.

## 5. Conclusions

Histoplasmosis is uncommonly diagnosed in Eastern Africa but should be considered in the differential diagnosis of an immunocompromised individual with unusual skin manifestations and persistent fevers unresponsive to standard antibiotics. Because the presentation of PDH can vary, histopathology should be obtained as part of the diagnostic workup. Despite the paucity of literature around outpatient amphotericin B administration, we report successfully treating an HIV positive child with PDH in an outpatient LMIC setting utilizing a multidisciplinary palliative care approach to maximize treatment success.

## Figures and Tables

**Figure 1 children-08-00273-f001:**
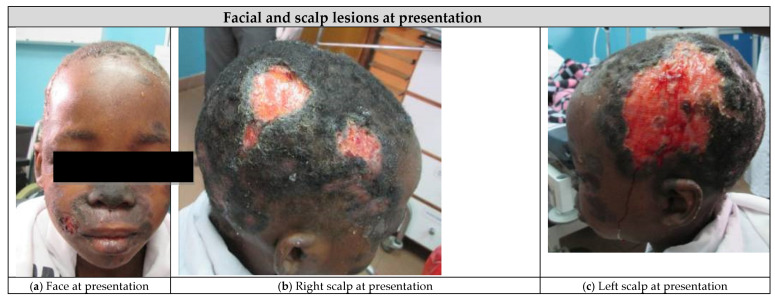
Initial presentation of histoplasmosis and response to antifungal therapy in a human immunodeficiency virus (HIV) positive child. (**a**) Face at presentation. (**b**) Right scalp at presentation. (**c**) Left scalp at presentation. (**d**) Face after amphotericin treatment (20 mg/kg). (**e**) Right scalp after amphotericin treatment (20 mg/kg). (**f**) Left scalp after amphotericin treatment (20 mg/kg). (**g**) Face at relapse. (**h**) Right scalp at relapse. (**i**) Left scalp at relapse. (**j**) Face after amphotericin treatment (39 mg/kg). (**k**) Right scalp after amphotericin treatment (39 mg/kg). (**l**) Left scalp after amphotericin treatment (39 mg/kg). (**m**) Face after 4 months of itraconzole treatment. (**n**) Right scalp after 4 months of itraconazole treatment. (**o**) Left scalp after 4 months of itraconazole treatment.

**Figure 2 children-08-00273-f002:**
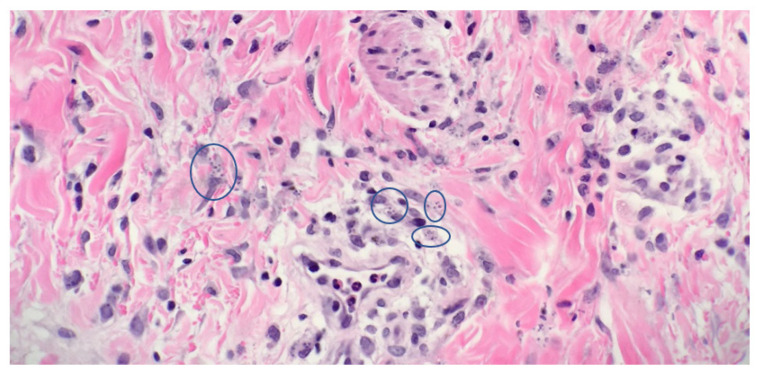
Histopathology showing within the superficial and the mid dermis; there is an interstitial infiltrate of lymphocytes, histiocytes, and neutrophils, with few eosinophils. Within the histiocytes, there are numerous organisms, consistent with histoplasmosis (Original magnification ×400).

**Figure 3 children-08-00273-f003:**
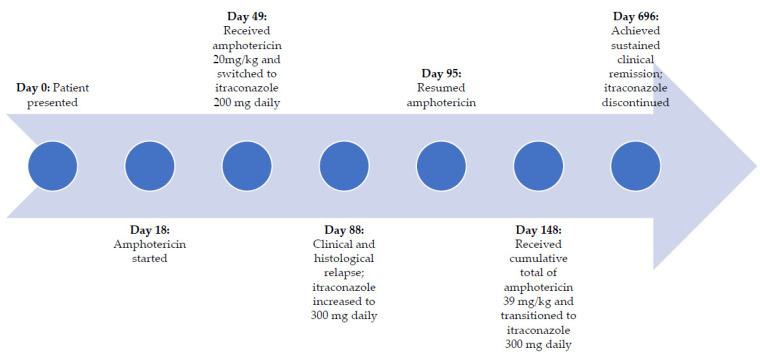
Timeline describing patient treatment and outcomes.

**Table 1 children-08-00273-t001:** Patient clinical and diagnostic findings at presentation.

Diagnostic Information	Result
Body mass index	12.1 kg/m^2 1^
Mid-upper arm circumference	12.5 cm ^2^
Absolute lymphocyte cell count	0.3 × 10^9^/L
Hemoglobin	2.98 mmol/L
CD4 count	24 cells/µL
HIV viral load	196,658 cp/mL
Chest radiograph	Bilateral infiltrates with mild hilar adenopathy
Skull radiograph	No abnormalities

^1^ This is greater than −3 standard deviations from the median. −3 WHO SD [5]. ^2^ For a child between the ages of 10–14 years, mid-upper arm circumference less than 16 cm is consistent with severe acute malnutrition [6]. CD4: cluster of differentiation 4 immune cells.

**Table 2 children-08-00273-t002:** Laboratory values prior to and after changes in patient’s antiretroviral regimen.

Laboratory Value	At Presentation ^1^	6 Months after Switch to ABC-3TC-LPV/r	12 Months after Switch to ABC-3TC-LPV/r	4 Years after Presentation; Treatment with ABC-3TC-DTG
CD4 count (cells/µL)	24	213	422	404
HIV viral load (cp/mL)	196,658	35	<20	<20
Hemoglobin (mmol/L)	2.98	5.52	6.14	9.06
White blood cell count (×10^9^/L)	3.43	4.63	5.5	No data available

ABC: abacavir; 3TC: lamivudine; LPV/r: lopinavir boosted with ritonavir; DTG: dolutegravir. ^1^ At presentation, the patient had received treatment with zidovudine, lamivudine, and efavirenz for 2 years.

## Data Availability

The datasets used and/or analyzed during the current study are available from the corresponding author on reasonable request.

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
