# Peer review of "Successful Use of Multidisciplinary Palliative Care in the Outpatient Treatment of Disseminated Histoplasmosis in an HIV Positive Child"

_children, 2021, doi:10.3390/children8040273_

Round 1

Reviewer 1 Report

This is a paper about the uncommon (in children) Histoplasmosis in an HIV positive child. The main question is about differential diagnosis of fever and disseminated skin ulcers in an HIV positive child. In my opinion, this is a very interesting case. The biopsy confirmed the result which makes it relevant. HIV positive status in children is becoming a chronic disease for readers in the Western and other non-African societies. However, this is not the case in Africa where HIV is still a cause of death and where opportunistic infections and cancers still occur in people living with HIV who are at AIDS stage. This is why this paper is interesting. The text is clear and well written, the presentation is easy to read. The multidisciplinary approach in children living with HIV is a must and is explained in details. This is the core of management in such cases and contexts and this has been proven in the manuscript. The conclusions are consistent with the importance of multidisciplinary approach and they address the main question. Furthermore, the usage of different drugs to treat Histoplasmosis is also a challenge in a child. The review of all cases in pediatric ages is also a plus. 

Reviewer 2 Report

This manuscript is based on an important topic and the authors have done a great job in reporting this case report. However, following a description of the major issues identified in this manuscript:

  1. Please provide a brief introduction of Histoplasmosis in HIV patients.
  2. In the section case report; Line no. 41-46 can be summarized in the table format. Please summarized in table format.
  3. Please provide in detail and timeline, how the patient was treated in a dedicated method section. Please provide a method section
  4. It would be interesting to compare these factors; CD4 counts, viral load, state of leukopenia, anemia, etc before and after switching to abacavir, lamivudine, and ritonavir-boosted lopinavir. Please include these details of these factors in a dedicated result section. Please provide a result section. Presentation of patient’s improvement in timeline format (as a figure file) will also help readers to understand easily.
  5. Please label the figure clearly as the face, scalp (left view/ right view), or as 1a, 1b, 1c, and so on at the time of presentation and also after the treatment. And explain briefly in the figure legend. It should be presented in such a way that just by looking at the figure and figure legends, it should be clear. Please revise.

Reviewer 3 Report

The authors preent an interesting case report with an unsual diagnosis and the reliance of palliative care to get the patient through the infectious episode.

My concern first is that there are 2 messages: 1/histoplasmosis is an uncommon diagnosis that should be kept in mind 2/the authros emphasize how they used palliative care to manage a complicated situation.

i understand there is a special issue on the topic of palliative care so i guess that is ok.

my main concern however is that i am skeptical about the diagnosis of "American" histoplasmosis. the cutaneous ulcerative lesions seem more in line with H. duboisii so the author should criticize the results from the lab perhaps ask them again with an clinical and pathology confrontation. were there any bone lesions?

it is said that it was disseminated histoplasmosis but the diagnosis was only performed on the skin where the any other organs that were positive for histo? this (disseminated or not) should be discussed

Round 2

Reviewer 2 Report

All concerns have been addressed in a satisfactory manner. 

thanks

Author Response

Thank you for the positive feedback and careful review.

Reviewer 3 Report

Although there has been improvement i am still in doubt about the final diagnosis 

I have shown the images to a professor of dermatology who is also a world expert on american histoplasmosis and he also has serious doubts.

The skin lesions presented look like duboisii or leishmaniasis (which responds to amphotericin b an can look similar under the microscope).

Has gomori grockott staining been performed?

There needs to be far more caution on the conclusions about the final diagnosis
